# Assessment of vaccination timeliness and associated factors among children in Toke Kutaye district, central Ethiopia: A Mixed study

**Kuma Dirirsa[1], Mulugeta Makuria[2], Ermias Mulu[2], Berhanu Senbeta Deriba[3]***

**1** Oromia Regional Health Bureau, Toke Kutaye Health Office, Guder, Ethiopia, **2** Department of Public Health, College of Medicine and Health Sciences, Ambo University, Ambo, Ethiopia, **3** Department of Public Health, Salale University College of Health Sciences, Fitche, Ethiopia

* berhanusenbeta55@gmail.com, berhanu_senbeta@slu.edu.et

**Data Availability Statement:** All relevant data are in the paper and its Supporting information files.

**Funding:** The authors received no specific funding for this work.

## Abstract

### Introduction

Age inappropriate vaccination of children increases the rate of mortality and morbidity. All studies conducted in some areas of Ethiopia were only quantitative in nature and focused on the main cities ignoring rural communities.

### Objective

The objective of this study is to assess vaccination timeliness and associated factors among children in Toke Kutaye district, central Ethiopia.

### Methods

A community-based cross-sectional study with quantitative and qualitative data collection methods was used, for which simple random sampling was used to select 602 mothers/caregivers who have vaccinated children aged 12 to 23 months in the district. The collected data were entered into Epi-data version 3.1 and exported to SPSS version 23 for analysis. Bivariate analysis with a P-value of < 0.25 was used to select candidate variables for multivariate logistic regression. Adjusted odds ratio (AOR) with 95% CI and p-value < 0.05 were used to declare a significant association. Qualitative data responses were classified and then organized by content with thematic analysis.

### Results

A total of 590 respondents responded to the interviews, making a response rate of 98%. In this study, 23.9% (95% CI: 20.4–27.7) of children aged 12–23 months had received all vaccines in the recommended time intervals. Urban residence (AOR: 3.15, 95% CI: 1.56–6.4), participation of pregnant women in conferences (AOR: 2.35, 95% CI: 1.2–4.57), institutional delivery (AOR: 2.5: 95% CI: 1.32–4.20), and sufficient knowledge of mothers (AOR: 3, 95% CI: 1.82–5.10) were significantly associated with the timeliness of childhood vaccination. Qualitative findings revealed that lack of knowledge and lack of information from mothers or caregivers, and inadequate communication with health workers hindered timely vaccination.

**Competing interests:** No author has a conflict of interest.

**Abbreviations:** ANC, Antenatal Care; BCG, Bacilli Chalmette Guerin; DHIS, District Health Information System; DPT, Diphtheria Pertussis Tetanus; EPI, Expanded Programs on Immunization; GVAP, Global Vaccine Action Plan; HCWs, Health Care Workers; HMIS, Health management information system; MNH, Maternal and Neonatal Health; PCV, Pneumococcal Conjugate Vaccine; PMTCT, Prevention of Mother to Child Transmission; PNC, Postnatal Care; SDGs, Sustainable Development Goals; UNICEF, United Nations Children's Fund; VPDs, Vaccine-Preventable Diseases and; WHO, World Health Organization.

## Conclusion

The overall timeliness of the child's vaccination was low in this study. Residence, participation in a conference, place of delivery, and knowledge of the mothers were predictors of vaccination timeliness. Hence, promoting institutional delivery and increasing pregnant mothers awreness on vaccination timeliness through conference participation is compulsory.

## Introduction

Immunization timeliness is the time at which child get a vaccine and by subtracting the infant's date of birth from the day of vaccination. Vaccinations were delivered on time if they were received within the World Health Organization (WHO) approved time frames and within first year of life [1]. Vaccine timeliness is critical in sub-Saharan African nations because vaccine-preventable diseases are important contributors to high child mortality [2]. The importance of timely infant immunization is to ensure that children have a good response, minimize individual vulnerability, and prevent disease outbreaks within communities [3]. Frequently, to assess the efficacy of an immunization system and population-level vulnerability, overall vaccination coverage or levels of vaccination for particular antigen by age group are used. However, simply measuring coverage does not provide information on whether vaccinations were administered on time or in line with the prescribed schedule [4]. Timely immunization is critical in the first year of life, since transplacental immunity decreases fast [5, 6].

Immunization coverage and age-appropriateness remain considerably lower in low-income nations than in middle- and high-income countries [7]. Glogally, 9 million children were died as a result of vaccine-preventable illness of which sub-Saharan Africa accounted for 4.4 million death in 2009 [8]. Vaccine doses administered before the recommended age or without respecting the dose interval can lead to suboptimal immune response [9, 10]. According to one study, infant vaccination doses were usually delayed, with 63.8 percent of Diphtheria Pertussis Tetanus (DTP) dose 1, 63.1 percent of Polio dose 1, and 68.5 percent of measles delivered more than one month after the recommended date in Ethiopia [11]. According to the Ethiopia Demographic and Health Survey (EDHS), data on vaccination coverage among children aged 12–23 months who received specific vaccines at any time prior to the survey revealed that only four out of ten children (43%) had received all basic vaccinations. The findings of the timeliness of child vaccination in EDHS 2014 and 2016, ranged from 18–22% [12, 13]. However, study finding from in Menz Lalo district, Amhara region, Ethiopia found that only 6.2% of the kids had received vaccinations at the right age [14]. According to a research done in Addis Abeba, Ethiopia reported that 55.94 percent of the children were vaccinations on time [15].

Most of the research in Ethiopia focused on Expanded Programs on Immunization (EPI) coverage [16–19]. However, high vaccination coverage rates for particular vaccines may not always reflect timely vaccination or population immunity. According to the annual Report of Toke Kutaye woreda Health Office, more than 85% of children received full immunization services, although the timeliness of vaccination status is unknown and not usually emphasized. This resulted in a missed opportunity to determine the prevalence of age-appropriate immunizations among children. Moreover, the previous study carried out in certain regions of Ethiopia were only quantitative in nature and focused on major cities. Even the previous quantitative studies found low vaccine timeliness that recommend the investigation of underlying factors. To close these gaps, this study tried to assess Vaccine timeliness and associated factors among children by applying mixed approach study including rural and urban residents.

## Methods and materials

### Study design, period, and setting

A community-based cross-sectional quantitative study was supplemented qualitative method. The study was carried out in Toke Kutaye District of West Shewa Zone, Central Ethiopia, from 1 May to 30 June 2020. The Toke Kutaye District is one of the 22 districts of the West Shewa Zone and located at 126 kilometers at west of Addis Ababa, Ethiopia's capital city. The district is separated administratively into four urban and twenty-three rural kebeles (the smallest administrative unit in Ethiopia). According to the woreda Health Office report, the Toke Kutaye district has a total population of 128,259, of whom 99,776 (78%) are rural residents, 28,483 (22%) are urban dwellers and a total of 26,721 households in 2019/2020. In these 2019/2020 fiscal years, the total population of children (12–23 months) was expected to be 3,724 (2.9 percent) [20].

### Source and study populations

All mothers or caregivers who had immunized infants aged 12 to 23 months who lived or resided in the Toke Kutaye district throughout the data collection period were source populations. All mothers/caregivers who had vaccinated children aged 12 to 23 months, lived in the specified kebeles at the time of data collection and included in the study were study populations. The study included mothers/caregivers with vaccinated children aged 12 to 23 months, who had lived in the study area for at least 6 months and were willing to participate.

### Sample size determinations

**For a quantitative study.**   The required sample size was calculated using a single population proportion formula by assuming a 95% confidence level, 5% margin of error. Where; Z = 1.96 with 95% confidence level, P = 39.1% (proportion of age-appropriate vaccine (timely) vaccine for pentavalent one from the study conducted in Menz Lalo district Northeast Ethiopia [14]. Taking into account the 10% non-response rate and the design effect 1.5, the final sample size was 602.

**For qualitative.**   Purposive sampling was used to select participants for the qualitative study.

Ten Focus Group Discussions (FGD) (one in each kebele) which comprise 8–12 individuals in each FGD were conducted with mothers who vaccinated their child from whom quantitative data were not collected. Participants of similar backgrounds in residents of the study area for more than six months were included. The modulators facilitated the FGD sessions, while the tape recorder recorded the responses of the FGDs respondents until the end of FGD discussion. The leaders of the kebele suggested the names of individuals who could participate in FGD. For the in-depth key informant interview (KII), three health extension workers (HEW) from three health posts and two nurses from two Health centers were selected purposely. Study interviews were conducted using a semi-structured interview guide with question-based questions related to the vaccine time lines aspect. We obtained the sample frame for this investigation by using a family folder.

### House hold family folder

The Family Folder is a pouch that is distributed to each home in the kebele. It provides information on the household that will assist the HEW in identifying the family or household's health (preventive, promotional, and environmental health) service requirements and providing the service or counseling accordingly. The Family Folder's front and back sides are used to

record information on: Household characteristics, latrine, hand-washing, waste disposal & drinking water facilities, and child health including vaccination status of children. The health cards and integrated maternal and child care cards are saved in the Family Folder for documenting illness information, preventative and promotional services to individual members of the household. Every family will get a Family Folder as part of government strategy to guarantee that every family receives family-centered health care. Health Extension Workers (HEWs) assigned to that health post (kebele) make house-to-house visits at least once a quarter to update household information while carrying a family folder. Vital registration data such as birth, migration, and death are updated daily based on reports from women's development armies and health extension workers on outreaches, however overall family folder data will be updated every quarter in accordance with government policy. Both immigration and emigration are reported by the Women's Development Army. Then, during outreach or home-to-home visits, health extension workers may confirm migration and record people who have moved into a new home in their field note book. Then they offer fresh family folders to immigrants and note on the family folder where they have moved for emigrants.

## Sampling procedure

The kebeles and the households were chosen using a multistage sampling approach. The district was divided into rural and urban Kebeles, and then 9 rural and 1 urban kebeles were drawn at random from a total of 4 urban and 23 rural kebeles in the district. A total of 13,11 children were estimated to be eligible in the selected kebeles. First, the list of all eligible children (13,11) aged 12 to 23 months was taken from all health posts family folders and vaccination records of selected kebeles' health posts. Furthermore, to ensure that no eligible kid was left out of the sampling frame, the list of eligible children obtained from the health posts family folder was cross-checked with the vaccination records in the health posts. In this manner, a comprehensive list of all eligible children in chosen 10 kebeles was created, comprising information such as a child's name, his/her parent's full name, the household's unique identification number, and the subkebele/got. Then proportional to size allocation was made to determine the required sample size from each kebele. Finally, a simple random sampling technique was used to select the required number of children from each kebele using the children listed as a sampling frame which was obtained from family folders. If eligible children's mother/care taker were not present at the time of data collection, a re-visit was arranged for a minimum of three times during the time of house hold survey. Purposive sampling was used to select participants (those whose children were vaccinated on time and those whose children were not vaccinated on time, urban dweller, and rural dwellers were included) for FGD conducted with mothers or caregivers and women development army group leaders and for key informant interviews (Fig 1).

## Variables

**Dependent variables.**   Childhood vaccination timeliness status.

## Independent variables

**Socio-Demographic Characteristics of the Children/Children's Mothers**: [Age, sex, marital status, number of children, residence, educational level, distance of Health Facility, mode of transport, occupation, telephone/mobile, and wealth index].

**Utilization of Maternal Health Services by Mothers**: [Participation in pregnant women conference (Pregnant women conference is a conference which is conducted at each kebele once per month by mid-wives and pregnant women to teach women about maternal and child

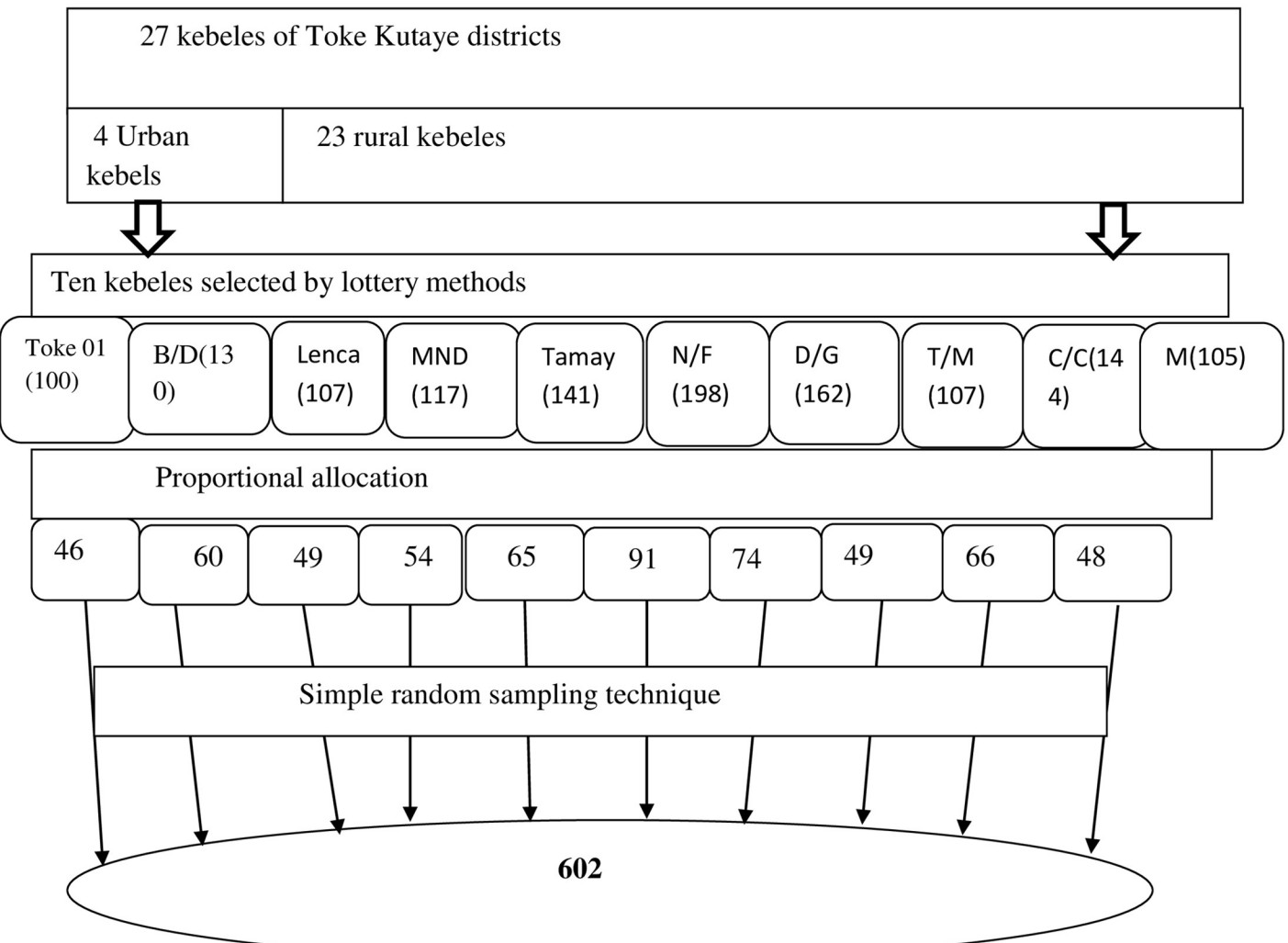

**Fig 1. Schematic presentation of a sampling technique for the timeliness of routine childhood vaccination and associated factors among children who vaccinated in the last year in Toke Kutaye District, West Shewa, Ethiopia, 2020.** Where B/D = Birbirsa Dogoma, MND = Melka nega denebe, N/F = Nega File, D/G = Deda Gelan, T/M = Toke Meti, C/C = Chancobi, and M = maruf.

health including vaccination), antenatal care utilization, place of delivery, post-natal care utilization, and receiving of tetanus toxoid vaccines, and season of birth of child].

**Awareness/knowledge of mother/caretaker about Vaccines and Vaccination**: [Knowledge of mother/caretaker on EPI information].

## Operational definitions and definition of terms

**Vaccine timeliness**: is the time at which child get a vaccine and by subtracting the infant's date of birth from the day of vaccination. Vaccinations were delivered on time if they were received within the World Health Organization (WHO) approved time frames [4] and checked by immunization card.

**Age-appropriate vaccination (timely)**: is measured if a child was vaccinated within one month after the minimum age to administer the dose as recommended by WHO.

**Table 1. Operational definition in relation to WHO & national vaccination schedule for respondents in Toke Kutaye District, West Shewa Zone, Oromia Region, Ethiopia, 2020.**

| Vaccine | WHO recommendation | | Operational definition | |
|---|---|---|---|---|
| | Minimum age | Minimum interval | Delayed | Early |
| BCG | 0 | 4 weeks | > 4 weeks | |
| OPV 1 | 6 weeks | 4 weeks | >10 week | <42 day |
| OPV 2 | 10 weeks | 4 weeks | >14 week | < 70 day |
| OPV 3 | 14 weeks | 4 weeks | >18 week | < 98 day |
| Pentavalent1 | 6 weeks | 4 weeks | >10 week | <42 day |
| Pentavalent2 | 10 weeks | 4 weeks | >14week | < 70 day |
| Pentavalent3 | 14 weeks | 4 weeks | >18 week | < 98 day |
| PCV1 | 6 weeks | 4 weeks | >10 week | <42 day |
| PCV 2 | 10 weeks | 4 weeks | >14week | < 70 day |
| PCV 3 | 14 weeks | 4 weeks | >18 weeks | < 98 day |
| Rota 1 | 6 weeks | 4 weeks | >14 week | < 42 day |
| Rota 2 | 10 weeks | 4 weeks | >18 week | < 70 day |
| Measles | 9 months | 4 weeks | > 10 months | < 270 days |

**Age-inappropriate vaccination (untimely)**: is measured if a child was vaccinated earlier and/or delayed than the recommended age.

**Delayed vaccination.** A vaccination is considered delayed if it is administered more than two weeks beyond the required age for BCG, polio, pentavalent, and PCV doses, or more than a month for measles.

**Early vaccination.** Any vaccination administered more than four days before the required age for each vaccine/dose was defined. Furthermore, for measles vaccination, we looked at doses administered more than two weeks before the recommended age (Table 1).

**Good knowledge.** Thirteen knowledge assessment item questions each containing (1 = yes and 0 = no) alternatives were used. From thirteen questions, women who answered seven or more questions correctly were considered as knowledgable whereas those who answered below seven were considered as not knowledgable.

## Data collection tool and techniques

Data were collected with face-to-face interviews using a structured questionnaire adapted from the Ethiopian Demographic Health Survey [13]. Age-appropriate vaccination schedule questions were developed from the WHO recommended schedule [4] and previously conducted similar research [14], which contain three parts. The first part was sociodemographic characteristics; the second part was the utilization of maternal health services by mothers, and the third part was awareness of mothers and barriers to vaccination service utilization related factors. The questionnaire was prepared in English and translated to Afan Oromo language by local language speakers who had BSc/Masters of Art in Afan Oromo language and the questionnaire was translated back to the English language by another individual who was blinded to the original English version and fluent in English and Afan Oromo, and comparison was made to check for its consistency. Finally, the Afan Oromo version was used to collect the data.

For qualitative data, open-ended guide questions for the FDGs and key informant interviews were developed in English and converted to Afan Oromo and then checked for validity. The tape recorder was used in the discussion and every discussion was recorded on the topics. Ten Bsc nurses and five BSC midwifery health professionals who were fluent Afan Oromo

speakers were recruited for data collection and supervision, respectively. The reliability of the questionnaires was checked with Cronbach's alpha with a value of 0.882 for vaccine timelines.

### Data quality control and management

The questionnaire was pre-tested on 5% (31women) of eligible women in Kolba Anchab kebele, which was not included in the study, and necessary modifications were made based on the nature of the identified gaps. Three days of training was provided to data collectors and supervisors. The investigators checked the completeness, precision, and consistency of all collected data every day. Data double entry was used to make comparisons of two data cells and determine if there was a difference.

### Data processing and analysis

The collected data were entered into the computer using Epi-data version 3.1 exported to SPSS version 23 for analysis. Logistic regression was used to identify important predictors of the timeliness of child vaccination. All covariates significant at p-value < 0.25 [21] in bivariate analysis were considered for further multivariate analysis. The fitness of the model was tested by the Hosmer-Lemeshow goodness-of-fit test. Finally, the adjusted odds ratio (AOR) with 95% CI and a p-value < 0.05 were used to declare a significant association.

**Qualitative data.** Qualitative data responses were categorized and then organized by content with thematic analysis. Data captured using tape records and notes was translated word by word into the English language and summarized manually in the main thematic area. Through this process, a verbatim was used to illustrate responses on relevant issues. The information obtained was triangulated with the quantitative information.

## Results

### Sociodemographic characteristics of the respondents

Of 602 mothers/caregivers, 590 responded to the interviews, making a response rate of 98%. Of all respondents, 572(97%) were mothers and 18(3%) were caregivers. The age of the mothers or caregivers in this study ranged from 14 to 45 years with a mean of 29.4±5.91 years. The wealth index result showed that 24.9% of the respondents were in the first quintile (poorest) (Table 2). This finding is supported by FGD discussants, as one 35-year-old mother explained: "... *health extension worker informs me about immunization to vaccinate my children because the health post is far from my house and I have no transportation cost, I was not vaccinated my child timely.*"

### Utilization of health institution by mother/caregiver

Approximately 101 (17.1%) of the children were born from an unplanned pregnancy and 27 percent of the mothers did not attend the conference of pregnant women at all; However, 370 (62.7%) of them had received at least three follow-ups of antenatal care. Concerning the place of delivery, 187(31.7%) had home delivery and 403 (67.3%) of the deliveries were in health facilities (Table 3).

### Awareness of vaccines and vaccine-preventable diseases

About one third of the mothers knew the time at which childhood vaccination started and 71.4% knew the age at which childhood vaccination ended correctly. Mothers or caregivers were asked to evaluate immunization services in their own opinion, given that in their residential area 62.2% of respondents mentioned that the delivery of immunization services is not too

**Table 2. Sociodemographic characteristics of the respondents in Toke Kutaye Woreda, West Shewa Zone, Oromia Region, Ethiopia, 2020 (N = 590).**

| Variables | Frequency | Percent |
|---|---|---|
| **Respondent** | | |
| Mother | 572 | 96.9 |
| Caregiver | 18 | 3.1 |
| **Age category of the mothers/caretakers** | | |
| <20 | 42 | 7.1 |
| 20–30 | 326 | 56.9 |
| 31–40 | 213 | 36.1 |
| ≥40 | 18 | 3.1 |
| **Sex of child** | | |
| Male | 315 | 53.4 |
| Female | 275 | 46.6 |
| **Family size** | | |
| < 5 | 507 | 85.9 |
| ≥ 5 | 83 | 14.1 |
| **Marital status of mother/care taker** | | |
| Married | 562 | 95.3 |
| Unmarried | 14 | 2.4 |
| Divorced | 5 | 0.8 |
| Widowed | 9 | 1.5 |
| **Residence** | | |
| Urban | 40 | 6.8 |
| Rural | 550 | 93.2 |
| **Mothers/caregivers educational status** | | |
| Unable to read and wright | 204 | 34.6 |
| Able to read and write | 175 | 29.7 |
| Only primary education | 152 | 25.8 |
| Secondary education | 49 | 80.3 |
| Diploma and above | 10 | 1.7 |
| **Mothers/caregivers occupation** | | |
| Farmer | 99 | 16.8 |
| House wife | 457 | 77.5 |
| Daily laborer | 12 | 2.0 |
| Government employee | 12 | 2.0 |
| Others[a] | 11 | 1.9 |
| **Mode of transportation** | | |
| On Foot | 529 | 89.5 |
| By Horse | 59 | 10.0 |
| By Car | 2 | 0.3 |
| **Distance of vaccination site** | | |
| Below 30 minutes | 169 | 28.6 |
| About30 minutes to one hour | 316 | 53.6 |
| Above one hour | 105 | 17.8 |
| **Wealth index/quintile** | | |
| Poorest | 147 | 24.9 |
| Poorer | 127 | 21.5 |
| Middle | 117 | 19.8 |

*(Continued)*

**Table 2.** (Continued)

| Variables | Frequency | Percent |
|---|---|---|
| Richer | 105 | 17.8 |
| Richest | 94 | 15.9 |

a = student, private work, and non-government organization.

bad in their residential area, about (32.4%) thought it was good, and 3.6% stated that they do not have any idea about the services, however 11 mothers or caregivers (1.9%) complained about the service. This finding is supported by FGD discussants, as a 26 years old women *who gave birth* explained: *"I go to the health post for the baby's vaccination at six weeks because I do*

**Table 3. Maternal health care practices of respondents in Toke Kutaye Woreda, West Shewa Zone, Oromia Region, Ethiopia, 2020 (N = 590).**

| Variables | Frequency | Percent |
|---|---|---|
| **Pregnancy status** | | |
| Planned | 489 | 82.9 |
| Unplanned | 101 | 17.1 |
| **Pregnant women's conference participation** | | |
| Not participated | 162 | 27.5 |
| ≤ 2 Participation | 346 | 58.6 |
| ≥ 3 Participation | 82 | 13.9 |
| **ANC follow up** | | |
| Yes | 520 | 88.1 |
| No | 70 | 11.9 |
| **Number of ANC visit** | | |
| ≤ 2 Participation | 150 | 25.4 |
| ≥ 3 Participation | 370 | 62.7 |
| **Place of delivery** | | |
| Home | 187 | 31.7 |
| Health facilities | 403 | 68.3 |
| **PNC follow up** | | |
| Yes | 326 | 55.3 |
| No | 264 | 44.7 |
| Number of PNC visit | | |
| one times | 233 | 39.5 |
| Two times | 90 | 15.3 |
| >2 | 3 | 0.5 |
| **TT status** | | |
| No dose received | 95 | 16.1 |
| 1 dose received | 106 | 18.0 |
| 2 dose received | 378 | 64.1 |
| >2 doses received | 11 | 1.9 |
| **Season of birth** | | |
| Summer | 222 | 37.6 |
| Winter | 123 | 20.8 |
| Autumn | 142 | 24.1 |
| Spring | 103 | 17.5 |

*not know the time at which child start vaccine.*" This finding is also supported by in-depth interview discussants: as one of the HEW discussant explained, "*. . . . . .I usually advise mothers on the correct time of immunization services, however, most mothers do not have better aware-ness about vaccination.*"

More than three-forth, 77.5% and 64,6% knew about vaccine preventable diseases such as poliomilitis and tuberculosis respectively (Table 4).

**Table 4. Awareness of mothers about vaccination utilization in Toke Kutaye District, West Shewa Zone Oromia, Ethiopia, 2020 (N = 590).**

| Variables | Frequency | Percent (%) |
|---|---|---|
| **EPI information** | | |
| Yes | 589 | 99.8 |
| No | 1 | 0.2 |
| **Source of EPI information** | | |
| Health profession | 325 | 55.1 |
| HEW | 571 | 96.8 |
| Radio | 175 | 29.7 |
| Friends | 24 | 4.1 |
| Neighbours | 4 | 1.0 |
| **know the benefit of vaccines** | | |
| Yes | 579 | 98.1 |
| No | 11 | 1.9 |
| **Benefit of vaccine meshed** (*) | | |
| To prevent the disease | 470 | 79.7 |
| For child health | 303 | 51.4 |
| For child groth | 2 | 0.3 |
| **List of vaccines by respondents** (*) | | |
| Tuberculosis | 381 | 64.6 |
| Poliomyelitis | 457 | 77.5 |
| Diphtheria | 267 | 45.3 |
| Pertussis | 192 | 32.5 |
| Diharrial | 251 | 42.5 |
| Measles | 430 | 72.9 |
| Tetanus | 194 | 32.9 |
| **know age at which child start vaccine** | | |
| Yes | 444 | 75.3 |
| No | 146 | 24.7 |
| **know age at which child ended vaccine** | | |
| Yes | 421 | 71.4 |
| No | 169 | 28.6 |
| **Immunization services status at health facility** | | |
| Good | 191 | 32.4 |
| Not too bad | 367 | 62.2 |
| Bad | 11 | 1.9 |
| No idea | 21 | 3.6 |
| **Knowledge of mothers** | | |
| Sufficient Knowledge | 381 | 64.6 |
| Insufficient Knowledge | 209 | 35.4 |

*More than one answer; percentages calculated among the total (n = 590).

**Table 5. Timeliness of vaccination among children aged 12–23 months in Toke Kutaye Woreda, West Shoa Zone, Oromia Region, Ethiopia, 2020.**

| Vaccination Schedule | Vaccine | Age appropriate time n (%) | Early n (%) | Delayed n (%) |
|---|---|---|---|---|
| Birth | BCG | 208(35.3) | 0 (0) | 382 (64.7 |
| 6 weeks | OPV 1 | 425(72.0) | 87(14.8) | 78 (13.2) |
| 10 weeks | OPV 2 | 409(69.3) | 63(10.7) | 118 (20.0) |
| 14 weeks | OPV 3 | 397(67.3) | 63(10.7) | 130 (22.0) |
| 6 weeks | PENTA 1 | 427(72.4) | 88 (14.9) | 75 (12.7) |
| 10 weeks | PENTA 2 | 413(70.0) | 69 (11.7) | 108 (18.3) |
| 14 weeks | PENTA 3 | 401(68.0) | 65 (11.0) | 124 (21.0) |
| 6 weeks | PCV 1 | 426(72.2) | 86 (14.6) | 78 (13.2) |
| 10 weeks | PCV 2 | 414(70.1) | 68 (11.6) | 108 (18.3) |
| 14 weeks | PCV 3 | 399(67.6) | 65 (11) | 126 (21.4) |
| 6 weeks | Rota 1 | 425(72.1) | 87 (14.8) | 77 (13.1) |
| 10 weeks | Rota 2 | 407(69.0) | 73 (12.4) | 110 (18.6) |
| 9 months | Measles | 260(44.1) | 176 (29.9) | 153 (26.0) |
| Overall timeliness | | 141(23.9) | 63 (10.7) | 386 (65.4) |

## Timeliness (early, age appropriate, and delayed) of child vaccination

Only 35.3% (95% CI: 32.7 to 40.8), 72.4% (95% CI: 68.5 to 75.9), 70% (95% CI: 66.1 to 73.6), 68% (95% CI: 64.1 to 71.7), and 44% (95% CI: 3936 to 48.4) were vaccinated at appropriate age for BCG, Pentavalent 1–3 and measles vaccine doses respectively. The proportion of antigens received earlier than the recommended national schedule was 14.9% (95% CI: 12.0 to 18.0), 11.7% (95% CI: 8.8 to 13.9), 11% (95% CI: 7.6 to 12.4) and 29.9% (95% CI: 29.4 to 48.4) for Pentavalent 1–3 and measles vaccine doses respectively. The magnitudes of delayed BCG, Pentavalent 1–3, and measles vaccination were 64.7% (95% CI: 59.2 to 67.3), 12.7% (95% CI: 10.0 to 15.6), 18.3% (95% CI: 15.9 to 22.5), 21% (95% CI: 19.0 to 25.4) and 26% (95% CI: 18.9 to 26.1) respectively. Timely vaccination was highest for Pentavalent one (72.4%) and lowest for BCG (35.3%) whereas untimely vaccination was highest for BCG (64.7%) and Measles (55.9%) compared to other vaccines in the EPI schedule. Overall, 23.9% (95%CI: 20.4–27.7) of children aged 12–23 months were received their vaccinations at the recommended time interval (Table 5).

## Reasons for age-inappropriate (not timely) immunizations

According to the survey findings, the majority (28.3%) of the respondents reported that the reasons for not following the appropriate time to receive vaccinations in a timely manner were lack of confirmed information. Among the respondents, one hundred forty-nine (25.3%) feared the adverse effect after immunization to follow the correct time to vaccinate their children (Fig 2).

This finding is supported by evidence from FGD discussants. The most frequently mentioned reasons as a barrier to age-appropriate vaccinations by the discussants were lack of information, the belief that it is not important unless the child feels sick, and the idea of attending the baby's immunization as important only for treatment. As one of the 29 years old discussants explained, "......*I gave birth at home... I did not know why I visit the hospital or the health center after delivery.*" (This is supported by others two FGD participants). Another 21 years old mother explained:, "*I do not consider returning back to the health facility is necessary after delivery and no one told me about Immunization service at the health facility, and when I have to return back to the health facility...*" (This is supported by others two FGD participants). The study participants perceived that attending EPI services was not important unless

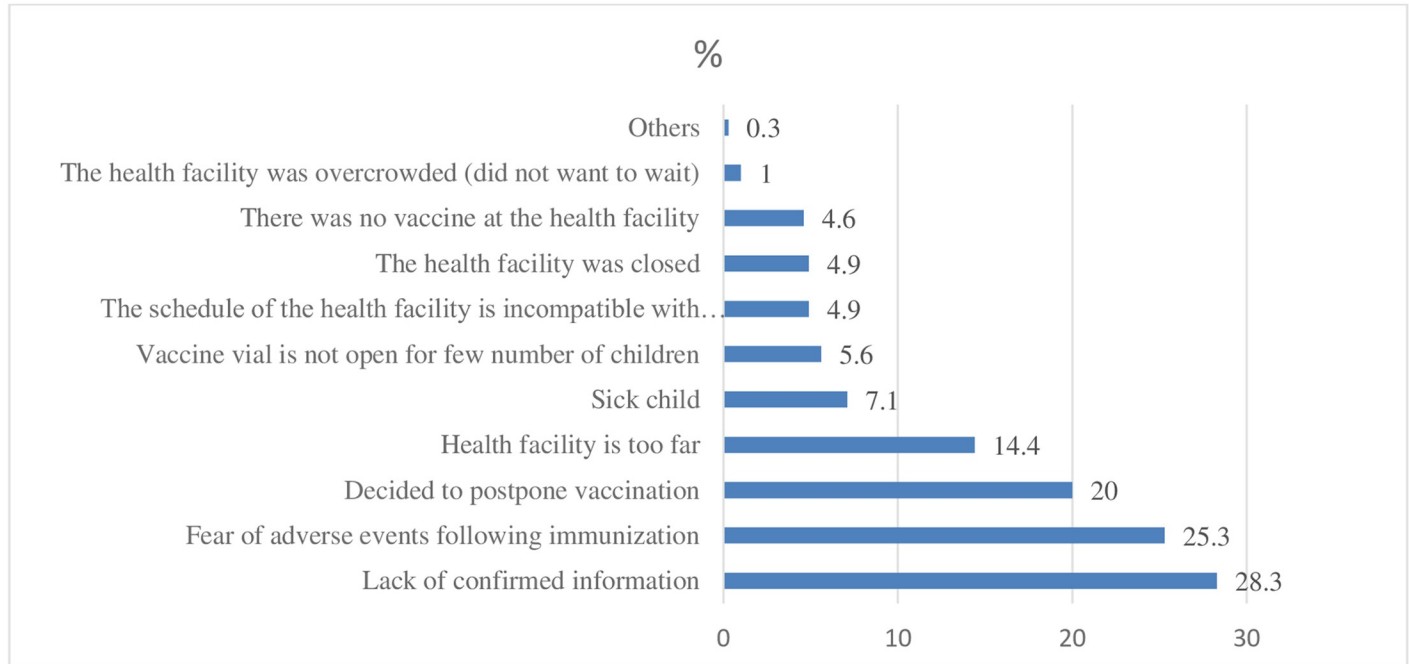

**Fig 2. Reason given by mothers or caregivers for not vaccinating their children on time in Toke Kutaye District, West Shewa Zone, 2020.**

their children were sick after delivery. A 36 years old woman FGD discussant explained, "...thanks to God, since I did not get sick and my child was fine, I did not go to the health facility for immunizations."

## Factors associated with childhood immunization timeliness

The age of mothers/caregivers, residence, educational status of the mother, participation in the pregnant women's conference, history of prenatal care follow-up (ANC), history of postnatal care follow-up (PNC), status of the Tetanus Toxoid (TT) vaccine, place of delivery, knowledge of the start time of vaccination, know the age at which the child reaches the vaccine, knowledge of the start time of vaccination and the end time, and knowledge of the mothers were the variables that showed a significant association with the timeliness of child vaccination in bivariate analysis.

The result of multivariate logistic regression analysis showed that children who lived in urban areas had 3.15 times higher odds of receiving vaccines at the recommended age compared to rural children (AOR: 3.15, 95% CI: 1.56–6.4). Children whose mothers participated in a pregnant women's conference had 2.35 times higher odds of receiving childhood vaccination in time compared to the odds of children whose mothers did not participate in the conference (AOR:2.35, 95% CI: 1.2–4.57). Mothers/caregivers who gave birth in Heath facilities had 2.5 times higher odds of vaccinating their children in time compared to those who gave birth at home (AOR: 2.5: 95% CI: 1.32–4.20). Mothers / caregivers who had sufficient knowledge of child vaccination had three times higher odds of taking their child's vaccination within the recommended time interval compared to the odds of their counterparts (AOR: 3, 95% CI: 1.82–5.10) (Table 6). This was supported by FGD as a 28 years old *mother said that "When mother took her child to the health facility, (for the health professionals) it is a good opportunity to give advice on the initiation time of the vaccine, when it should get sated age appropriate time and importance of timely completion of immunization for child so having follow-up for service*

**Table 6. Factors associated with vaccination timeliness among children aged 12–23 months in Toke Kutaye District, West Shewa Zone, Oromia, Ethiopia, 2020 (N = 590).**

| Variables | Timeliness of vaccinations | | COR 95% CI | AOR 95% CI |
|---|---|---|---|---|
| | Age appropriate | Age inappropriate | | |
| **Residence** | | | | |
| Urban | 22(3.7%) | 18(3.0%) | 4.40(2.3–8.50)* | 3.15(1.566.4)** |
| Rural | 119(20.2%) | 430(73.0) | 1 | 1 |
| **Pregnant women's conference participation** | | | | |
| Yes | 128(21.7%) | 300(50.9%) | 4.48(2.5–8.0)* | 2.35(1.2–4.57)** |
| No | 13(2.2%) | 148(25.1%) | 1 | 1 |
| **Place of delivery** | | | | |
| Home | 17(2.9%) | 170(28.9%) | 1 | 1 |
| Health facilities | 124(21.1%) | 278(47.2%) | 4.50(2.6–7.7)* | 2.50(1.32–4.2)** |
| **Knowledge of Mothers** | | | | |
| Sufficient Knowledge | 119(20.2%) | 262(44.5%) | 3.80(2.35–6.3)* | 3.0(1.82–5.10)** |
| Insufficient Knowledge | 22(3.7%) | 186(31.6%) | 1 | 1 |

COR = Crude Odds Ratio

* = P<0.25; AOR = Adjusted Odds Ratio

** = Significant level at p-value <0.05, 1 = reference

*utilization of the health institution is necessary for all mothers.*" (This idea was supported by another 36 years woman).

## Discussion

According to the findings of this study, overall childhood vaccination timeliness was 23.9 percent among children aged 12 to 23 months. This means that the remaining 76.1 percent of children remained vulnerable to vaccine-preventable diseases, as failing to be vaccinated on time would lengthen children's susceptibility period by reducing herd immunity. This figure is higher than the 6.2 percent discovered in the Menz Lalo region of northeast Ethiopia [14], and the 6.1 percent discovered in Kenya [11]. This mismatch might be related to differences in socioeconomic factors, study duration, design, site, and health-care access. This figure is comparable to the Ethiopia DHS 2016 finding of 22% [13]. This consistency might be attributed to similar study settings and the usage of EPI services throughout the country. This study result is low compared to Tanzania's 41 percent [22], Nigeria's 48 percent [23], Kampala, Uganda's 45.6 percent [24], and China's 44 percent [25]. This discrepancy might be explained by differences in study participant characteristics, sociodemographic factors, access to health care, and less or no attention to vaccination timeliness, as well as the lack of a timeliness indicator in Ethiopia's immunization program and including only vaccinated children in the current study. This study discovered a gap in the timeliness of children's immunization. Timely vaccinations of each antigen was low when compared to total: BCG (36.6 percent), OPV1 (72 percent), OPV2 (69.3 percent), OPV3 (67.3 percent), Pentavalent1 (72.4 percent), Pentavalent2 (70 percent), Pentavalent 3 (68 percent), PCV1 (72.1 percent), PCV2 (70 percent), PCV3 (67.6 percent), Rota1 (72 percent), Rota 2 (69.2 percent) and Measles (44 percent) were vaccinated at a suitable age. The finding of this study is higher compared to the findings in Menz Lalo district, Northeast Ethiopia for Penta1 (39.1%), Penta 2 (36.3%), Penta 3 (30.3%), and Measles (26.4%) (14). This study is also high compared to the study in Nigeria Penta 1 (61.5%), Penta2 (51.7%) and Penta 3 (46.7%) [23] and in China BCG (30%) penta1 (28%) and Measles (12%)

[25]. This discrepancy might be due to differences in study participant characteristics, study period, design, and health service accessibility. The finding of this study is lower compared to studies conducted in Senegal, BCG (88.25%), Penta1 (74%), Penta 2 (75.64%) [26], Gambia, BCG (94.3%), Penta1 (78%), OPV1 (74.6%), and Measles (80%) [27]. This could be because the national vaccination policy, which has been aimed at achieving more than 95% vaccination coverage in 2030, had simply focused on up-to-date coverage, irrespective of the time of vaccination [3]. In general, a higher percentage of mothers vaccinated their children 96% to 100% for all antigens in the district, it is only 23.9% of mothers that met the required number of vaccinated their children as recommended by WHO. This clearly indicates that the recommended utilization of the EPI service in the district is still poor.

Giving Birth at the health institution was positively associated with vaccine timelines in this study. This result was supported by studies done in the Menz Lalo area of northeast Ethiopia [14], Dessie town [17], and Gambia [27], that indicated that giving birth in a health facility had a direct and substantial connection with vaccine timeliness. This is because mothers who gave birth at a health facility had a better chance of being informed and receiving health education about the benefits of EPI services. This enhances child health care in general, including the behavior of moms seeking childhood vaccinations. In the current study, being an urban resident was an independent predictor of the child's vaccination timeliness. This finding is consistent with one of Ethiopian findings [14]. This might be because urban resident moms have better information and realize the significance of vaccination, as well as variations in the availability, accessibility, and functionality of health services in urban areas than rural areas. In the current study, children of mothers / caregivers who attended pregnant women's conferences were more likely to vaccinate their children at the proper age than those who did not attend the conferences. This result is consistent with findings from earlier research in the Menz Lalo area of Northeast Ethiopia [14]. This is due to the moms who attended pregnant women conferences may have received more information, understanding about timely immunization of their children, greater awareness about vaccine-preventable illnesses, and recognized the value of vaccines. Children of mothers/caregivers who had sufficient knowledge about childhood vaccination were more likely to be immunized at the recommended time interval than children of mothers/caregivers who had insufficient knowledge of vaccines and VPDs. This finding is consistent with research done in the Menz Lalo region of North East Ethiopia [14] and in Dessie Town [17]. The possible reason is because moms or caregivers who understand the childhood vaccination schedule, vaccine-preventable illnesses, and reasons for vaccination are more likely to bring their kids to the immunization location at the appropriate time. This finding is supported by the evidence from FGD participant. The most commonly stated barriers to age-appropriate vaccination were a lack of knowledge, the belief that it is not necessary unless the mother is unwell, and viewing attendance to EPI services as vital primarily for the care of the infant. As one participant put it, "I didn't know when, how many times, or why I go to the health center for immunization."

## Strength of the study

Being a community-based study and utilizing qualitative research to examine areas that the quantitative survey did not address.

## Limitation of the study

The study's participants were chosen based on the presence of vaccination cards, which may have resulted in selection bias because infants whose parents did not keep their immunization cards were excluded. Using family folder to get study population which may has poor quality

sice it is not electronic based. Since this study included only vaccinated children, the proportion of untimely vaccination was slightly larger and unable to evaluate vaccine coverage. Another limitation of this study is unable to include children more than 23 months old who could receive vaccination in the future time.

## Conclusion

This study conclude that the overall vaccination timeliness of children was poor. Residence, attending the conference of pregnant women, site of delivery, and knowledge of mothers were factors associated with the timeliness of vaccination. As a result, health leaders and policy-makers should pay attention to and incorporate vaccination timeliness into the EPI program. Furthermore, promoting institutional delivery and increasing pregnant mothers awreness on vaccination timeliness through conference participation is compulsory to improve childhood vaccination timeliness.

## Supporting information

**S1 File.**
(RAR)

## Acknowledgments

We thank Ambo University's College of Medicine and Health Sciences, Department of Public Health, for their assistance in ensuring the success of this research. Our heartfelt gratitude also goes to district administrators, data collectors, supervisors, and all research participants for their valuable assistance in completing the study.

## Author Contributions

**Conceptualization:** Kuma Dirirsa, Mulugeta Makuria, Ermias Mulu, Berhanu Senbeta Deriba.

**Data curation:** Kuma Dirirsa, Mulugeta Makuria, Ermias Mulu.

**Formal analysis:** Kuma Dirirsa, Mulugeta Makuria, Ermias Mulu, Berhanu Senbeta Deriba.

**Funding acquisition:** Kuma Dirirsa, Mulugeta Makuria, Ermias Mulu, Berhanu Senbeta Deriba.

**Investigation:** Kuma Dirirsa, Mulugeta Makuria, Ermias Mulu, Berhanu Senbeta Deriba.

**Methodology:** Kuma Dirirsa, Mulugeta Makuria, Ermias Mulu, Berhanu Senbeta Deriba.

**Project administration:** Kuma Dirirsa, Mulugeta Makuria, Ermias Mulu, Berhanu Senbeta Deriba.

**Resources:** Kuma Dirirsa, Mulugeta Makuria, Ermias Mulu.

**Software:** Kuma Dirirsa, Mulugeta Makuria, Ermias Mulu, Berhanu Senbeta Deriba.

**Supervision:** Kuma Dirirsa, Mulugeta Makuria, Ermias Mulu, Berhanu Senbeta Deriba.

**Validation:** Kuma Dirirsa, Mulugeta Makuria, Ermias Mulu, Berhanu Senbeta Deriba.

**Visualization:** Kuma Dirirsa, Mulugeta Makuria, Ermias Mulu, Berhanu Senbeta Deriba.

**Writing – original draft:** Kuma Dirirsa, Mulugeta Makuria, Ermias Mulu, Berhanu Senbeta Deriba.

**Writing – review & editing:** Kuma Dirirsa, Mulugeta Makuria, Ermias Mulu, Berhanu Senbeta Deriba.

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
