## [Decision Letter · Decision Letter 0]

9 Sep 2021

PONE-D-20-40561Assessment of vaccination timeliness and associated factors among Children in Toke Kutaye District, Central Ethiopia: A Community based cross-sectional study

PLOS ONE

Dear Dr. Deriba,

Thank you for submitting your manuscript to PLOS ONE. After careful consideration, we feel that it has merit but does not fully meet PLOS ONE’s publication criteria as it currently stands. Therefore, we invite you to submit a revised version of the manuscript that addresses the points raised during the review process.

We look forward to receiving your revised manuscript.

Kind regards,

Bidhubhusan Mahapatra, Ph.D.

Academic Editor

PLOS ONE

Additional Editor Comments (if provided):

This is very important work on immunization and can be good contribution if the authors can address the comments provided by both the reviewers. In addition to the reviewers' suggestions, please elaborate the qualitative research piece with more clarity: how many FGDs were conducted? How it was done? How the coding was done? What software was used for analysis? Please also re-work on the measure section to clearly define all variables included in the analysis irrespective of whether they are outcomes or predictors. Finally, I suggest you take the service of an English language editor as there are several misspelling, and grammatically errors in the sentences. I also advise authors to examine the PLOS One author guidelines and structure your paper accordingly.

Journal Requirements:

2. Please include additional information regarding the survey or questionnaire used in the study and ensure that you have provided sufficient details that others could replicate the analyses. For instance, if you developed a questionnaire as part of this study and it is not under a copyright more restrictive than CC-BY, please include a copy, in both the original language and English, as Supporting Information.  If the original language is written in non-Latin characters, for example Amharic, Chinese, or Korean, please use a file format that ensures these characters are visible.

3. Please state whether you validated the questionnaire prior to testing on study participants. Please provide details regarding the validation group within the methods section.

4. Please include a copy of the interview guide used in the study, in both the original language and English, as Supporting Information, or include a citation if it has been published previously.

5. Thank you for stating the following financial disclosure: "No"

6. We note that you have indicated that data from this study are available upon request. PLOS only allows data to be available upon request if there are legal or ethical restrictions on sharing data publicly. For more information on unacceptable data access restrictions, please see http://journals.plos.org/plosone/s/data-availability#loc-unacceptable-data-access-restrictions. 

7. Please amend your list of authors on the manuscript to ensure that each author is linked to an affiliation. Authors’ affiliations should reflect the institution where the work was done (if authors moved subsequently, you can also list the new affiliation stating “current affiliation:….” as necessary).

8. We noticed you have some minor occurrence of overlapping text with the following previous publication(s), which needs to be addressed:

- https://www.researchsquare.com/article/rs-56134/v1

- https://panafrican-med-journal.com/content/series/27/3/8/full/

- https://www.tandfonline.com/doi/full/10.1080/21645515.2018.1480242

- http://tropmedhealth.biomedcentral.com/articles/10.1186/s41182-016-0013-x

The text that needs to be addressed involves the Introduction and parts of the Discussion section. 

In your revision ensure you cite all your sources (including your own works), and quote or rephrase any duplicated text outside the methods section. Further consideration is dependent on these concerns being addressed.

Reviewers' comments:

Reviewer's Responses to Questions

**Comments to the Author**

1. Is the manuscript technically sound, and do the data support the conclusions?

Reviewer #1: Partly

Reviewer #2: Yes

2. Has the statistical analysis been performed appropriately and rigorously? 

Reviewer #1: Yes

Reviewer #2: Yes

3. Have the authors made all data underlying the findings in their manuscript fully available?

Reviewer #1: No

Reviewer #2: Yes

4. Is the manuscript presented in an intelligible fashion and written in standard English?

Reviewer #1: Yes

Reviewer #2: No

5. Review Comments to the Author

Reviewer #1: This paper has to do with a very important subject topic in public health related to timeliness of routine vaccination and associated factors. Though the WHO fixes the minimum age required to be eligible for various vaccines in the Expanded Program on Immunization (EPI), it is important to note that the minimum age is based on the evidence or believe that vaccinating the child before the said age might to induce the desired immune response but does not in away mean that the children are not at risk of the various vaccine preventable diseases. It is therefore a well justified topic to merit consideration viewed it potential public health significance. The authors applied a mixed methods of study i.e. community based cross sectional study design coupled with FGD and informants’ interview. The methodologic approach is quite sound given that it is a complex topic and requires critical examination from various dimensions.

However, well will like to make some comments that if addressed might improve the quality of the paper.

Abstract

1. Methods part of the abstract fails to describe briefing the sampling technique for the quantitative component of the study. We suggest this should be included.

2. The results section should give a brief description of the sample composition. For instance, how many children were included in total, what proportion of children receive their scheduled vaccine etc. In as far as the timeliness is important, herd immunity is created in the community by high vaccination coverage. Late vaccination poses an individual risk while low coverage poses but individual and community risk.

3. The first sentence says “In this study, 23.9% (95%CI: 20.4 -27.7) of children aged 12-23 months had received all vaccines at the recommended time intervals”. It is importance to clearly define “recommended time interval” in this context. The EPI objective is that a child completes all his/her vaccines before 12 months (0-11months) but it is not clear if this is what the sentence means or something else. Please clarify.

Introduction

1. Statement like “Out of 9 million deaths of children globally as a result of vaccine-preventabledisease, a bigger proportion occurred in sub-Saharan Africa which was 4.4 million” requires the author to specify the year. It is important for the reader to know when the situation was reported.

2. The sentence “Vaccine doses administered too close together or too early can lead to suboptimal immune response” needs to be rephrase. In fact, for the vaccines in the EPI, we do not have counter indication of administering them together. It only poses a problem of doses of the same vaccine are administered without respective the recommended 4 weeks interval. On the other hand, what is meant by “too early”? I suggest the sentence should be “vaccine doses administered before the recommended age or without respecting the dose-interval can lead to suboptimal immune response”

Methods

1. In general, this study relied on the records of households at the level of the health facility. This is a bit risky as we do not have a clou on how the households’ folders are updated and managed. Are the household folders electronic or paper register? How is the update done? When updating the folders, how is population movement managed? Has any study been done to assess the quality of data in these folders? It will be good to open a subheading under materials and methods to write about the households’ folders and provide the answers to all up mentioned question. The population is very dynamic, and therefore some children in the family folders might have left to a different woreda where they can still complete their immunization and on time while others not born in the woreda, who had started or even completed their vaccination schedule in a different woreda may arrive at various ages. These “new arrival” may or may not have their vaccination card. How were such cases treated in the family folders at health facility? The understanding of these question is paramount to a better understanding of the designs and its potential limitations in this study.

2. One issue with external validity of this study is that the investigators seem to have included only children vaccinated. In fact, children who did not receive any vaccine at all forms part of those who did not receive their vaccines on time. That is why it was important for the team not to assess timeliness without assessing coverage. Please kindly clarify the study population.

3. For the qualitative section of this study, the investigators said that only family who had resided in the kebeles for at least 6 months were included in the study. Please justify this decision.

4. Furthermore, you said the kebeles header suggested informant that could give reliable information. What do you mean by “reliable information”? I would have preferred a FGD and interview of various families purposely selecting those whose children where vaccinated on time and those whose children were not vaccinated on time urban dweller and rural dwellers. In purposive sampling, there is still some logical selection of people. It will be good to describe the people included finally in the results section so that it will guide the interpretation of the results.

5. What is the meaning of “If the eligible children were not present during the visit, a revisit was arranged at a minimum of three times during the time of the survey”? where was the data sources, health facility vaccination register or the households? This need to be clarified in the paper. It is unclear where the researchers had the household folders and health facility vaccination register but choose to collect data with interview at households on children immunization. Did they think that the registers may not be up to date?

Results

1. The results section should bring out the vaccination coverage before checking the timeliness

Reviewer #2: The authors provide a manuscript that explores factors associated with vaccination timeliness among children in Ethiopia. Given the (relatively) recent push to improve vaccination timeliness alongside vaccination coverage, the topic of this manuscript is timely. To improve the impact of the paper, I would encourage the authors to seek out a manuscript editing service, as many portions of the paper were difficult to understand (in terms of grammar, punctuation, and word choice). More specific comments are below:

ABSTRACT

-There is a typo in the results section. You have written that "insufficient knowledge of mothers" was associated with vaccination timeliness, but in the body of the paper you say that more vaccination knowledge is associated with timeliness.

INTRODUCTION

-The authors should more specifically define what vaccination timeliness is in the introduction. Please also provide a more detailed explanation of the importance of vaccination timeliness (versus simple vaccination coverage) and why it is important to understand the factors associated with vaccination timeliness.

-It would be helpful for the audience (who may not be well versed in vaccination schedules) for you to briefly list out which vaccines are included in the Ethiopian EPI vs what the ‘basic’ vaccinations are before citing statistics about vaccination coverage.

-Please fully write out a what an acronym stands for before using the acronym for the first time. For example, in the introduction you use acronyms for each of the vaccine doses and for EPI but you do not define what the acronyms stand for. The list of acronyms at the end of the paper is helpful but is not a substitute for defining the acronyms in the paper.

METHODS

-2.1: Write out what FGD stands for before using the acronym.

-2.4: Include the total number of urban and rural kebeles in the district (i.e. 4 urban and 23 rural).

-2.4: What is the difference between the 3,742 eligible children and the 1,311 eligible children? Please clarify.

-2.5: For maternal knowledge - Why were women who scored greater than the mean number of questions considered to have good knowledge? A more robust measure of maternal knowledge would be to determine a set number of questions a woman must answer correctly to be considered knowledgeable, rather than simply correctly answering more questions than the average woman in the sample.

-2.7: Provide the number of women (not just the percentage) who completed the pre-test.

-2.8: Please include a list of all predictors included in bivariate analyses as well as their categories. Also, please describe (briefly) what a pregnant woman’s conference is for those who are not familiar with the Ethiopian context.

RESULTS

-3.3: Include results of awareness of vaccine-preventable disease.

-3.4: Include proportions of children with early and delayed vaccination. Also, report overall coverage of vaccines, to contrast differences in general coverage of a vaccine and timely uptake of a vaccine.

-3.5: You provide direct quotes and state that it is quoted material from multiple women – please report the quote from the woman who directly stated it and then mention the number of women who agreed with her comment.

-3.6: Define what TT is before using the abbreviation.

-3.6: The second paragraph starts with “After checking of confounding variables…” If you are examining predictors of timely vaccination (rather than examining a specific relationship between a particular exposure and timely vaccination) you would not have confounders, as there is no relationship to confound. If there were covariates you included in the model (whose relationship to timely vaccination is not being tested) then you should list out those covariates in the methods section.

DISCUSSION

-The authors should more fully discuss the implications of the results and potential reasons for the relationships between timely vaccination and associated factors beyond stating the findings are consistent with previous literature.

-Please expand upon both the strengths and the limitations of the analysis. Consider how aspects of the study design, sampling method, topic area, locale, methods, etc. are strengths/limitations of the study. Please expand upon your comment about how only including children with vaccination cards would influence your results. Additionally, you mentioned in the methods (2.2) that children had to be vaccinated to be eligible for the study. If unvaccinated children were not included in your study, the proportion of untimely vaccination is likely larger than you report (unvaccinated children can be thought of as children with long delays in their vaccination, as they have not yet received a vaccine by 12-23 months but could still receive it in the future). Please discuss this limitation.

REFERENCES

-Review references and ensure they have all relevant fields and are formatted correctly. For example, references 3 and 8 refer to the World Health Organization as “Organization WH.” Additionally some references (ex. #5) do not have journal titles included. Finally, reference 3 and 26 are duplicates.

TABLES AND FIGURES

-Overall, review the tables for consistency (i.e., use the same number of decimals for all percentages) and spelling/grammar.

-Figure 1: What are the numbers in parentheses under each of the kebeles and why do some of them have letters (ex. MND (117) and Maru f(105))? Clarify in a footnote.

-Figure 1: What are the numbers under “proportional allocation?” Clarify in a footnote.

-Figure 2: I would recommend converting Figure 2 into a table that also includes general coverage of each vaccine dose among children 12-23 months. This would allow you to explicitly compare vaccination coverage vs timeliness. Plus, the figure is difficult to read in black and white (the age appropriate and early colors look like the same color) and the numbers on the figure overlap, making them difficult to read.

-Figure 3: Label the x-axis (percent) rather than titling the figure “%”

6. PLOS authors have the option to publish the peer review history of their article (what does this mean?). If published, this will include your full peer review and any attached files.

Reviewer #1: No

Reviewer #2: No

---

## [Author Response · Author response to Decision Letter 0]

8 Nov 2021

PONE-D-20-40561

Title: Assessment of vaccination timeliness and associated factors among Children in Toke Kutaye District, Central Ethiopia: A mixed study

To: Editorial Office of PLO ONE

Dear Sir or Madam. 

As it is to be recalled, I have sent a paper entitled “Assessment of vaccination timeliness and associated factors among Children in Toke Kutaye District, Central Ethiopia: A mixed study” to be published on your journal. First of all, we would like to thank the reviewers and editorials for their valuable and constructive comments for our manuscript. Thus, as per the reviewers and editorials comments we have revised and modified the document and also we tried to address each comment one by one. We hope the revised manuscript addressed the reviewer’s concerns and has resulted in a paper that is clearer, and more persuasive. We also want to confirm you that this manuscript is not submitted elsewhere and had only been submitted to the current Journal.

With regards

 Berhanu Senbeta Deriba (Lecturer, Researcher, MPH)

 Corresponding author

Dear reviewers 

We thank you for reviewing and providing us with important and valid comments that helped us to enrich our manuscript. Really we would like to appreciate and thank you for your valuable, constructive comments too. Based on your suggestions, we have incorporated the comments into the manuscript, and we have also provided a response to each comment as follows. 

Additional Editor Comments (if provided):

This is very important work on immunization and can be good contribution if the authors can address the comments provided by both the reviewers. In addition to the reviewers' suggestions, please elaborate the qualitative research piece with more clarity: how many FGDs were conducted? 

Author Response: Ten FGDs were conducted.

How it was done? 

Author Response: Participants of similar backgrounds in residents of the study area for more than six months were included. The modulators facilitated, the FGDs sessions while the tape recorder recorded the responses of the FGDs respondents until the end of FGD discussion. The kebele leaders suggested the names of individuals who could participate in FGD. The study interviews were held using a semi structured interview guide with probing questions linked to the vaccine time lines aspect.

How the coding was done? 

Author Response: The transcripts were examined by the research team prior to coding to identify significant themes and build a code book. All members of the study team coded the transcripts. To increase inter-coder reliability, the coders utilized the code book separately, and coding discrepancies were resolved through conversation. Following that, a final edition of the code book was created, as well as categories and topics. For each topic and category, the coded transcripts were further examined and summarized in narratives.

What software was used for analysis? 

Author Response: It was analyzed manually and no software was used for analysis of qualitative data.

Please also re-work on the measure section to clearly define all variables included in the analysis irrespective of whether they are outcomes or predictors. Finally, I suggest you take the service of an English language editor as there are several misspelling, and grammatically errors in the sentences. I also advise authors to examine the PLOS One author guidelines and structure your paper accordingly.

Author Response: Thank you; we have corrected it.

Journal Requirements:

Author Response: Thank you we have corrected it.

2. Please include additional information regarding the survey or questionnaire used in the study and ensure that you have provided sufficient details that others could replicate the analyses. For instance, if you developed a questionnaire as part of this study and it is not under a copyright more restrictive than CC-BY, please include a copy, in both the original language and English, as Supporting Information. If the original language is written in non-Latin characters, for example Amharic, Chinese, or Korean, please use a file format that ensures these characters are visible.

Author Response: Thank you; we have made correction.

3. Please state whether you validated the questionnaire prior to testing on study participants. Please provide details regarding the validation group within the methods section.

Author Response: Thank you for your constructive comment; we have accepted the comments and corrected it accordingly.

4. Please include a copy of the interview guide used in the study, in both the original language and English, as Supporting Information, or include a citation if it has been published previously.

Author Response: Thank you for your constructive comment; we have accepted the comments and corrected it accordingly.

5. Thank you for stating the following financial disclosure: "No"

Author Response: Thank you for your constructive comment; we have accepted the comments and corrected it accordingly. The authors received no specific funding for this work.

6. We note that you have indicated that data from this study are available upon request. PLOS only allows data to be available upon request if there are legal or ethical restrictions on sharing data publicly. For more information on unacceptable data access restrictions, please see http://journals.plos.org/plosone/s/data-availability#loc-unacceptable-data-access-restrictions. 

Author Response: Thank you for your constructive comment; we have accepted the comments and corrected it accordingly.

7. Please amend your list of authors on the manuscript to ensure that each author is linked to an affiliation. Authors’ affiliations should reflect the institution where the work was done (if authors moved subsequently, you can also list the new affiliation stating “current affiliation:.” as necessary).

Author Response: No author was moved from his former organization. Therefore, keep the affiliation of authors as it was stated in the manuscript before.

8. We noticed you have some minor occurrence of overlapping text with the following previous publication(s), which needs to be addressed:

- https://www.researchsquare.com/article/rs-56134/v1

- https://panafrican-med-journal.com/content/series/27/3/8/full/

- https://www.tandfonline.com/doi/full/10.1080/21645515.2018.1480242

- http://tropmedhealth.biomedcentral.com/articles/10.1186/s41182-016-0013-x

The text that needs to be addressed involves the Introduction and parts of the Discussion section. 

In your revision ensure you cite all your sources (including your own works), and quote or rephrase any duplicated text outside the methods section. Further consideration is dependent on these concerns being addressed. 

Author Response: Thank you for your constructive comment; we have accepted the comments and corrected it accordingly.

Reviewers' comments:

Reviewer's Responses to Questions

Comments to the Author

1. Is the manuscript technically sound, and do the data support the conclusions?

Reviewer #1: Partly

Reviewer #2: Yes

2. Has the statistical analysis been performed appropriately and rigorously?

Reviewer #1: Yes

Reviewer #2: Yes

3. Have the authors made all data underlying the findings in their manuscript fully available?

Reviewer #1: No

Reviewer #2: Yes 

Author Response: Thank you for your constructive comment; we have accepted the comments and corrected it accordingly.

4. Is the manuscript presented in an intelligible fashion and written in standard English?

Reviewer #1: Yes

Reviewer #2: No

Author Response: We would like to appreciate your valuable comments. As a result, we have accepted the comments and corrected it accordingly.

5. Review Comments to the Author

Reviewer #1: This paper has to do with a very important subject topic in public health related to timeliness of routine vaccination and associated factors. Though the WHO fixes the minimum age required to be eligible for various vaccines in the Expanded Program on Immunization (EPI), it is important to note that the minimum age is based on the evidence or believe that vaccinating the child before the said age might to induce the desired immune response but does not in away mean that the children are not at risk of the various vaccine preventable diseases. It is therefore a well justified topic to merit consideration viewed it potential public health significance. The authors applied a mixed methods of study i.e. community based cross sectional study design coupled with FGD and informants’ interview. The methodologic approach is quite sound given that it is a complex topic and requires critical examination from various dimensions.

However, well will like to make some comments that if addressed might improve the quality of the paper.

Abstract

1. Methods part of the abstract fails to describe briefing the sampling technique for the quantitative component of the study. We suggest this should be included.

Authors Response: Thank you for your nice and constructive comments; we have accepted the comment and corrected it accordingly.

2. The results section should give a brief description of the sample composition. For instance, how many children were included in total, what proportion of children receive their scheduled vaccine etc. In as far as the timeliness is important, herd immunity is created in the community by high vaccination coverage. Late vaccination poses an individual risk while low coverage poses but individual and community risk.

Authors Response: We thank you for your valuable comment. We have accepted the comment and corrected it.

3. The first sentence says “In this study, 23.9% (95%CI: 20.4 -27.7) of children aged 12-23 months had received all vaccines at the recommended time intervals”. It is importance to clearly define “recommended time interval” in this context. The EPI objective is that a child completes all his/her vaccines before 12 months (0-11months) but it is not clear if this is what the sentence means or something else. Please clarify.

Authors Response: It means age appropriate vaccination. Age-appropriate vaccination (timely): is measured if a child was vaccinated within one month after the minimum age to administer the dose as recommended by WHO.

Introduction

1. Statement like “Out of 9 million deaths of children globally as a result of vaccine-preventable disease, a bigger proportion occurred in sub-Saharan Africa which was 4.4 million” requires the author to specify the year. It is important for the reader to know when the situation was reported.

Authors Response: Thank you we have corrected it accordingly.

2. The sentence “Vaccine doses administered too close together or too early can lead to suboptimal immune response” needs to be rephrase. In fact, for the vaccines in the EPI, we do not have counter indication of administering them together. It only poses a problem of doses of the same vaccine are administered without respective the recommended 4 weeks’ interval. On the other hand, what is meant by “too early”?

Authors Response: Early vaccination: Any vaccination administered more than four days before the required age for each vaccine/dose was defined. Furthermore, for the measles vaccination, we looked at doses administered more than two weeks before the recommended age.

 I suggest the sentence should be “vaccine doses administered before the recommended age or without respecting the dose-interval can lead to suboptimal immune response”

Authors Response: It is a nice observation so that we have accepted the comments and corrected it accordingly.

Methods

1. In general, this study relied on the records of households at the level of the health facility. This is a bit risky as we do not have a clue on how the households’ folders are updated and managed. Are the household folders electronic or paper register? 

Authors Response: Paper register, but updated regularly.

How is the update done?

Authors Response: Vital registration like birth, migration and death are updated every day by the report of women development army and health extension workers out reach works, but overall family folder data will be updated every quarter as the government policy.

 When updating the folders, how is population movement managed?

Authors Response: As mentioned above, one way women development army report both immigration and emigration. Another way during out rich or home to home visit health extension workers can see migration or those who changed house register on their field note book. Then they give new family folder for immigrants and mention on family folder where they have moved for emigrants on their family folder.

 Has any study been done to assess the quality of data in these folders? 

Authors Response: Yes

It will be good to open a subheading under materials and methods to write about the households’ folders and provide the answers to all up mentioned question. The population is very dynamic, and therefore some children in the family folders might have left to a different woreda where they can still complete their immunization and on time while others not born in the woreda, who had started or even completed their vaccination schedule in a different woreda may arrive at various ages. These “new arrival” may or may not have their vaccination card. How were such cases treated in the family folders at health facility? 

Authors Response: We have corrected it accordingly.

The understanding of these question is paramount to a better understanding of the designs and its potential limitations in this study.

Authors Response: We have include this issue in the limitation of the study.

2. One issue with external validity of this study is that the investigators seem to have included only children vaccinated. In fact, children who did not receive any vaccine at all forms part of those who did not receive their vaccines on time. That is why it was important for the team not to assess timeliness without assessing coverage. Please kindly clarify the study population.

Authors Response: All mothers and caregivers who had immunized infants aged 12 to 23 months who lived or resided in the Toke Kutaye district throughout the data collection period were considered source populations. Mothers and caregivers who had no vaccinated children were not source population for this study. 

3. For the qualitative section of this study, the investigators said that only family who had resided in the kebeles for at least 6 months were included in the study. Please justify this decision.

Authors Response: This is to consider permanent residents of the district and not to include those who came from another area because they cannot represent the district in which the study was carried out. Moreover, those who did not live in the district for less than six months do not have family folder from which the evidence of the family was obtained.

4. Furthermore, you said the kebeles header suggested informant that could give reliable information. What do you mean by “reliable information”?

Authors Response: We have corrected it because it created certain sort of confusion. What we mean by reliable information was that those who explain themselves and about EPI well for qualitative study.

 I would have preferred a FGD and interview of various families purposely selecting those whose children were vaccinated on time and those whose children were not vaccinated on time urban dweller and rural dwellers. In purposive sampling, there is still some logical selection of people. It will be good to describe the people included finally in the results section so that it will guide the interpretation of the results.

Authors Response: Thank for your valuable comments’ we have accepted the comments and corrected it accordingly.

5. What is the meaning of “If the eligible children were not present during the visit, a revisit was arranged at a minimum of three times during the time of the survey”? where was the data sources, health facility vaccination register or the households? This need to be clarified in the paper.

Authors Response: We used family folders available at health posts for obtaining lists of vaccinated children (sampling frame). The data were collected at community level at the households. Any time the child start vaccine the mother or care taker will be given immunization card which will be filled immunization status at every immunization visit. 

It is unclear where the researchers had the household folders and health facility vaccination register but choose to collect data with interview at households on children immunization. Did they think that the registers may not be up to date?

Authors Response: The registration and family folder only has the evidence of time when child took vaccine, but they do not have full information about factors associated with vaccine timelines and others relevant data for the study.

Results

1. The results section should bring out the vaccination coverage before checking the timeliness

Authors Response: We did not collected data from unvaccinated children and it is not in lined with our study objectives.

Reviewer #2: The authors provide a manuscript that explores factors associated with vaccination timeliness among children in Ethiopia. Given the (relatively) recent push to improve vaccination timeliness alongside vaccination coverage, the topic of this manuscript is timely. To improve the impact of the paper, I would encourage the authors to seek out a manuscript editing service, as many portions of the paper were difficult to understand (in terms of grammar, punctuation, and word choice). More specific comments are below:

Authors Response: Thank you for your nice observation; we have corrected it accordingly.

ABSTRACT

-There is a typo in the results section. You have written that "insufficient knowledge of mothers" was associated with vaccination timeliness, but in the body of the paper you say that more vaccination knowledge is associated with timeliness.

Authors Response: Thank you for your nice observation; we have corrected it accordingly.

INTRODUCTION

-The authors should more specifically define what vaccination timeliness is in the introduction. Please also provide a more detailed explanation of the importance of vaccination timeliness (versus simple vaccination coverage) and why it is important to understand the factors associated with vaccination timeliness.

Authors Response: We thank you for your nice observation. We have accepted the comments and corrected it in the revised manuscript

-It would be helpful for the audience (who may not be well versed in vaccination schedules) for you to briefly list out which vaccines are included in the Ethiopian EPI vs what the ‘basic’ vaccinations are before citing statistics about vaccination coverage.

Authors Response: Thank you for your constructive comment. We have mentioned this issue under operational definition of vaccination timeliness. Would you see table 1 please?

-Please fully write out a what an acronym stands for before using the acronym for the first time. For example, in the introduction you use acronyms for each of the vaccine doses and for EPI but you do not define what the acronyms stand for. The list of acronyms at the end of the paper is helpful but is not a substitute for defining the acronyms in the paper.

Authors Response: Thank you for your nice observation; we have corrected it accordingly.

METHODS

-2.1: Write out what FGD stands for before using the acronym.

Authors Response: Thank you for your nice observation; we have corrected it accordingly.

-2.4: Include the total number of urban and rural kebeles in the district (i.e. 4 urban and 23 rural).

Authors Response: Thank you, we have corrected it accordingly.

-2.4: What is the difference between the 3,742 eligible children and the 1,311 eligible children? Please clarify.

Authors Response: Thank you, we have corrected it accordingly.

It was edition error: actually 3,742 was the total number of 12-23 months old children in the district ( 27 kebeles) and 1,311 was 12-23 months old children in the selected kebeles (10 kebeles).

-2.5: For maternal knowledge - Why were women who scored greater than the mean number of questions considered to have good knowledge? A more robust measure of maternal knowledge would be to determine a set number of questions a woman must answer correctly to be considered knowledgeable, rather than simply correctly answering more questions than the average woman in the sample.

Authors Response: Thank you for your valuable comment, we have corrected it accordingly.

-2.7: Provide the number of women (not just the percentage) who completed the pre-test.

Authors Response: Thank you; We have corrected it accordingly.

-2.8: Please include a list of all predictors included in bivariate analyses as well as their categories. 

Authors Response: Thank you; We have corrected it accordingly.

Also, please describe (briefly) what a pregnant woman’s conference is for those who are not familiar with the Ethiopian context.

Authors Response: Pregnant women conference is a conference which is conducted at each kebele once per month by mid-wives and pregnant women to teach women about maternal and child health including vaccination.

RESULTS

-3.3: Include results of awareness of vaccine-preventable disease.

Authors Response: Thank you; we have corrected it accordingly.

-3.4: Include proportions of children with early and delayed vaccination. Also, report overall coverage of vaccines, to contrast differences in general coverage of a vaccine and timely uptake of a vaccine.

Authors Response: Thank you for your valuable comment, we have corrected it accordingly.

-3.5: You provide direct quotes and state that it is quoted material from multiple women – please report the quote from the woman who directly stated it and then mention the number of women who agreed with her comment.

Authors Response: Thank you for your valuable comment, we have corrected it accordingly.

-3.6: Define what TT is before using the abbreviation.

Authors Response: We have corrected it accordingly.

-3.6: The second paragraph starts with “After checking of confounding variables…” If you are examining predictors of timely vaccination (rather than examining a specific relationship between a particular exposure and timely vaccination) you would not have confounders, as there is no relationship to confound. If there were covariates you included in the model (whose relationship to timely vaccination is not being tested) then you should list out those covariates in the methods section.

Authors Response: Thank you for your valuable comment, we have corrected it accordingly.

DISCUSSION

-The authors should more fully discuss the implications of the results and potential reasons for the relationships between timely vaccination and associated factors beyond stating the findings are consistent with previous literature.

Authors Response: Thank you; we have corrected it accordingly.

-Please expand upon both the strengths and the limitations of the analysis. Consider how aspects of the study design, sampling method, topic area, locale, methods, etc. are strengths/limitations of the study. Please expand upon your comment about how only including children with vaccination cards would influence your results.

Authors Response: Thank you for your valuable comment, we have corrected it accordingly.

 Additionally, you mentioned in the methods (2.2) that children had to be vaccinated to be eligible for the study. If unvaccinated children were not included in your study, the proportion of untimely vaccination is likely larger than you report (unvaccinated children can be thought of as children with long delays in their vaccination, as they have not yet received a vaccine by 12-23 months but could still receive it in the future). Please discuss this limitation.

Authors Response: Thank you; it is a nice observation and we have corrected it accordingly.

REFERENCES

-Review references and ensure they have all relevant fields and are formatted correctly. For example, references 3 and 8 refer to the World Health Organization as “Organization WH.” 

Authors Response: Thank you; We have corrected it accordingly.

Additionally, some references (ex. #5) do not have journal titles included. Finally, reference 3 and 26 are duplicates.

Authors Response: It is unpublished on Journal but released on line.

TABLES AND FIGURES

-Overall, review the tables for consistency (i.e., use the same number of decimals for all percentages) and spelling/grammar.

Authors Response: 

-Figure 1: What are the numbers in parentheses under each of the kebeles and why do some of them have letters (ex. MND (117) and Maruf(105))? Clarify in a footnote.

Figure 1: What are the numbers under “proportional allocation?” Clarify in a footnote.

Authors Response: We have corrected accordingly.

-Figure 2: I would recommend converting Figure 2 into a table that also includes general coverage of each vaccine dose among children 12-23 months. This would allow you to explicitly compare vaccination coverage vs timeliness. Plus, the figure is difficult to read in black and white (the age appropriate and early colors look like the same color) and the numbers on the figure overlap, making them difficult to read.

Authors Response: We have made the correction.

-Figure 3: Label the x-axis (percent) rather than titling the figure “%”

Authors Response: We have agreed with your comment, but majority of the texts in the graph are long, we are unable to correct it as per your comment.

---

## [Decision Letter · Decision Letter 1]

23 Dec 2021

Assessment of vaccination timeliness and associated factors among children in  Toke Kutaye district, central Ethiopia: A Mixed study.

PONE-D-20-40561R1

Dear Dr. Deriba,

We’re pleased to inform you that your manuscript has been judged scientifically suitable for publication and will be formally accepted for publication once it meets all outstanding technical requirements.

Kind regards,

Bidhubhusan Mahapatra, Ph.D.

Academic Editor

PLOS ONE

Additional Editor Comments (optional):

Reviewers' comments:

Reviewer's Responses to Questions

**Comments to the Author**

1. If the authors have adequately addressed your comments raised in a previous round of review and you feel that this manuscript is now acceptable for publication, you may indicate that here to bypass the “Comments to the Author” section, enter your conflict of interest statement in the “Confidential to Editor” section, and submit your "Accept" recommendation.

Reviewer #2: All comments have been addressed

2. Is the manuscript technically sound, and do the data support the conclusions?

Reviewer #2: Yes

3. Has the statistical analysis been performed appropriately and rigorously? 

Reviewer #2: Yes

4. Have the authors made all data underlying the findings in their manuscript fully available?

Reviewer #2: Yes

5. Is the manuscript presented in an intelligible fashion and written in standard English?

Reviewer #2: Yes

6. Review Comments to the Author

Reviewer #2: (No Response)

7. PLOS authors have the option to publish the peer review history of their article (what does this mean?). If published, this will include your full peer review and any attached files.

Reviewer #2: No

---

## [Editor Report · Acceptance letter]

19 Jan 2022

PONE-D-20-40561R1 

Assessment of vaccination timeliness and associated factors among children in  Toke Kutaye district, central Ethiopia: A Mixed study. 

Dear Dr. Deriba:

I'm pleased to inform you that your manuscript has been deemed suitable for publication in PLOS ONE. Congratulations! Your manuscript is now with our production department. 

Kind regards, 

on behalf of

Dr. Bidhubhusan Mahapatra 

Academic Editor

PLOS ONE